# Beyond coincidence: An investigation of the interplay between synchronicity awareness and the mindful state

**Eyal Rosenstreich** [1,2] *, **Pninit Russo-Netzer** [3], **Tamar Ickeson** [1,4]

1 School of Behavioral Sciences, Peres Academic Center, Rehovot, Israel, 2 School of Human Movement and Sport Sciences, The Levinsky-Wingate Academic College, Netanya, Israel, 3 Faculty of Education and Leadership, and Department of Psychology, Achva Academic College, Arugot, Israel, 4 Departments of Management and School of Education, Ben-Gurion University, Beer-Sheva, Israel

* eyal@pac.ac.il

**Data Availability Statement:** https://osf.io/v5fw6/?view_only=314a1f1809dc42f684a8f899c3049d85.

**Funding:** The author(s) received no specific funding for this work.

## Abstract

The mindful state is commonly characterized by an elevated awareness of the present moment. An intriguing and rather widespread phenomenon that requires the attention to the present moment is Synchronicity Awareness. This phenomenon refers to the sense of a profound psychological connection between an internal event (e.g., thought, image, or dream) and external events. Whereas mindfulness and its underlying cognitive processes have been well documented, synchronicity awareness, despite its deep roots, has been scarcely examined empirically, and little is known about the cognitive mechanisms underlying it. The present study aimed to further validate the recently developed Synchronicity Awareness and Meaning Detection (SAMD) scale and explore its potential relationship with various mindfulness measures. To this end, 572 participants engaged in an online survey, incorporating the SAMD, Five Facets of Mindfulness Questionnaire, Mindful Awareness and Attention Scale, Langer's Mindfulness Scale, and Encoding Style Questionnaire. The results revealed that synchronicity awareness, meaning detection, and mindfulness are distinct constructs. A significant association between synchronicity and core facets of mindfulness was observed, indicating that participants with higher SAMD scores were more sensitive to inner sensations, more novelty-seeking, and engaged in their actions. Additionally, SAMD scores were linked to internal encoding style, suggesting a top-down processing of experiences. However, synchronicity was not associated with outwards-directed mindful awareness, suggesting that it might involve an intrinsic attentional process, influenced by internal cues. Theoretical and practical implications are discussed.

## Introduction

Mindfulness is commonly defined as an elevated state of awareness to the present moment. This mindful state is characterized in increased attention directed both towards internal sensations and external experiences, while maintaining a curious and non-judgmental perspective

**Competing interests:** The authors have declared that no competing interests exist.

[1, 2]. In this study we aim to bring into focus a yet to be explored aspect of the mindful state, *Synchronicity Awareness* [3, 4].

Mindfulness has been conceived as an acquired skill [5, 6] and as a trait-like construct [1]. Although some scholars have suggested mindfulness to be a single trait, such as the tendency to be attentive and aware of distal and proximal stimuli [7], there is a consensus that mindfulness is better conceived as a multifaceted construct, entailing not only the awareness of stimuli but also their 'untainted' perception [8]. Such mindfulness facets are the attentional factor itself (interoceptive sensitivity and awareness towards external stimuli) and the qualities of the present moment experiences (maintaining a nonjudgmental and nonreactive stances, being curious and open towards them, and being able to describe them) [1, 2, 8]. Similarly, some scholars see mindfulness as the ability to notice new elements in the surroundings and the production of new and innovative thought about them [9]. Such meta-cognitive approach to mindfulness defines the mindful-state as increased sensitivity to the environment, openness to new information, promoting new perspectives, and acknowledging other points of view of the situation at hand [10].

Previous studies have demonstrated that mindfulness may be associated with altered processing of information. For example, mindfulness has been argued to be associated with the processing of information at the fringes of consciousness, which manifested with increased sensitivity to metacognitive processing and evoking 'a feeling of memory' during memory task performance [11, 12]. In line with this notion, mindfulness has been argued to be associated with more creative thinking among organization employees [13]. Finally, mindfulness was found to be associated with an external encoding style, which is best conceived as a rather slow, bottom-up, processing of information and more resilient to cognitive errors [14].

Taken together, it seems that mindfulness enables the incorporation of both holistic processing, in which a large amount and rather generalized information is employed for maintaining wide perspective, and analytic processing, in which the object or the situation at hand is delved into. This study explores the interplay between these concepts and a deeply rooted, yet rather new, empirically explored area of synchronicity experiences.

## Mindfulness and synchronicity awareness

Jung [15] initially defined synchronicity as a profound psychological connection between an internal event (such as a thought, image, or dream) and one or more external events that occur simultaneously. Essentially, these experiences surpass mere chance and embody "the coincidence of events in space and time as meaning something more" [16]. Hocoy further elaborated on the term, stating that it combines the Greek words for "joined with" and "in time," implying a bond that forms through temporal correspondence [17]. According to the Jungian perspective, synchronicity embodies a holistic encounter in which the significance of an external event becomes apparent when it aligns with an individual's inner realm [16]. This interdependence between one's internal and external experiences hinges upon the subjective interpretation of events.

However, despite its deep roots in early psychological theory, there has been a significant lack of systematic scientific investigation into synchronicity experiences as a common everyday experience. The amount of empirical research conducted on this concept has been scarce, with the majority of studies focused on psychotherapy and career development. To address this gap, a phenomenological analysis using in-depth interviews was conducted [3], yielding an integrative heuristic model characterizing such experiences as involving Receptiveness-Exceptional encounter-Meaning-detecting (REM). These elements include receptiveness (R), which involves increased attention and openness to both one's internal and external world, serving as a prerequisite for an exceptional encounter (E). Exceptional encounter refers to

unexpected event that aligns with an individual's inner state of mind, evoking distinct and memorable emotions. Lastly, meaning-detecting (M) refers to a conscious process of linking the event to one's personal life narrative, where participants derived meaning from their synchronicity experiences by uncovering a hidden order, an organizing framework that imbued their lives with comprehensibility and a belief that experiences and events carry significance.

The precondition to experiencing synchronicity, as identified by the participants—increased intention and paying attention to the environment—echoes established concepts within the mindfulness literature in relation to the significance of receptive attention to the present moment [18] and awareness focused on current experiences [7]. Such a mindset appears to enhance individuals' ability to cope with uncertainty [19] and to mitigate defensive responses to threats [20].

Beyond such potential correspondence between these lines of research, synchronicity experiences further emphasize the importance of maintaining an open and receptive mindset toward unexpected and unexplained experiences in life [3, 4]. Cognitive studies have indicated that people incorporate chance signals into a holistic, well-structured, and genuine narrative of their own lives, enabling them to view the world as logical, purposeful, and inherently positive [21, 22]. Experiences of synchronicity require being attuned to random, unexpected external cues. At the same time, individuals are required to remain connected to their inner, unconscious world. This unique context, with its complex environment and multiple stimuli, provides fertile ground for exploring mindfulness concepts.

The present study is a first attempt to explore complementary and distinct aspects of mindfulness concepts and synchronicity experiences. For this purpose, we used the SAMD scale [4], a validated measure that was created based on the REM model [3] to assess individual differences in the experience of synchronicity with regard to both aspects of the experience (i.e., awareness and meaning-detecting). A previous study found that synchronicity awareness and meaning-detecting are positively correlated with openness to experience and tolerance for ambiguity. It was also found that individuals who actively search for meaning, remain open to synchronicity events, and successfully make sense of them tend to experience greater meaning and optimism, which ultimately contributes to higher life satisfaction [4]. In sum, it may be suggested that being aware of our present-moment experiences enhances our ability to facilitate a dialogue between our internal world and external environment and to make sense of random and unexpected information and events in individuals' everyday lives in natural settings.

The aim of the present study was thus twofold: (1) to further validate the structure of the SAMD scale by replicating its results in an independent and international sample; and (2) to examine the interplay between synchronicity awareness and related concepts of mindfulness, in order to better understand the unique contributions of such ingredients. We predicted that SAMD scores will be overall associated with higher mindfulness scores, in particular with its attentional factors (interoceptive sensitivity and awareness towards external stimuli). Second, we predicted that SAMD scores will be positively associated with the ability to notice new elements in the surroundings and the production of new and innovative thoughts about them (i.e., the Langerian mindfulness). Finally, SAMD scores were predicted to be associated with an internal encoding style.

## Materials and methods

### Participants

To estimate a-priori required sample size, we used G-power software with the following statistical assumptions: type 1 error of 5% and minimum statistical power of 95% [23]. Based on

previous studies, moderate associations were expected between the study variables [4]. For this reason, minimum sample size was estimated at 266 respondents. The overall sample consisted of 572 participants from the USA and the UK. Participant's ages ranged from 18 to 80 with mean age of 42.95 years (*SD* = 14.03). Approximately 50% of the sample (*n = 285*) identified as male, 49.5% identified as female (*n = 283*) and the rest identified as none or didn't report their gender. Number of education years ranged between 10 to 35 with average of 15.41 years (*SD* = 3.46). 95.26% of the sample were Caucasian. As for relationship status, 71% reported they were in a relationship, 29% reported they were not. As for religiosity level, 53.5% reported they were not religious at all, while less than 1% reported they were very religious. Participants were compensated with a payment of 1.5 pounds sterling (about 1.8 USD), which equals to about 7 pounds sterling per hour.

## Procedure

All data was collected between February 15–16, 2022. Participants were recruited through Prolific (online participant recruitment platform; www.Prolific.co) and received monetary compensation. Recently, online panels have become a common way to target and reach respondents in social science research [24, 25]. The prolific panel was found to have better quality than other recruitment platforms (e.g., Amazon Mechanical Turk) [26, 27]. All participants completed a series of online questionnaires. Prior to filling out the questionnaires, participants provided a signed informed consent, which specified the purpose of the research, its procedures, and the voluntary nature of participation. Participants were guaranteed anonymity, and no disclosure of personal details was required. The study was approved by the IRB in the first author's academic institution (IRB number 2021–380).

## Measures

**Synchronicity Awareness** was assessed with the Synchronicity Awareness and Meaning-Detecting (SAMD) Scale [4.] The SAMD scale is comprised of two subscales: (a) synchronicity awareness (SA), and (b) synchronicity meaning-detecting (MD). Since the first dimension refers to the *frequency* of specific events and the second dimension to the *subjective perception* of such events, a different response format (i.e., 6- and 7-point scales) and different number of items are adopted for each dimension.

The SA subscale refers to awareness of the occurrence of synchronicity events in daily lives. It involves 9 items using a 6-point scale (0 = never, 1 = once, 2 = twice or more, 3 = rarely, 4 = often, 5 = all the time). An example item is: "I ran into something or someone that I thought about in an unexpected place". Internal reliabilities in the current study were good: Cronbach's $\alpha$ and McDonald's ω coefficients were .81. As for the prevalence of synchronicity experiences, 92.13% to all participants reported they experienced at least one or more such encounters.

The MD subscale refers to the meaning detected in the synchronicity events or experiences. It involves 13 items using a 7-point scale (1 = not at all to 7 = to a high degree). An example item is: "I believe that unexplained events enable new discovery and development". Internal reliabilities in the current study were good: Cronbach's $\alpha$ and McDonald's ω coefficients were .91.

**Dispositional trait mindfulness** was assessed using two common instruments. One instrument was the shortened form of the five facets of mindfulness questionnaire (FFMQSF) [28]. The five facets of dispositional mindfulness are assessed by measuring: nonreactivity (the ability to withhold thoughts, emotional expressions, and physical actions), ability to observe (the direction of attention to the surroundings), acting with awareness (being aware of inner and sensations and outer stimuli), ability to describe (the ability to verbalize thoughts and feelings),

and being non-judgmental (accepting the self and the others as they are). It's 24 statements (e.g., "I notice the smells and aromas of things") were rated on a 6-point Likert scale (1 = "never or very rarely true", 5 = "very often or always true"). Items of two facets, awareness and being non-judgmental, were presented in reveres scales, such that high score corresponded with being less aware and more judgmental; these scales were transformed before the analysis of the data to correspond with the direction of the other scales. Regarding the internal reliabilities of the subscales: nonreactivity subscale had Cronbach's $\alpha$ and McDonald's ω coefficients of .85, ability to observe subscale had Cronbach's $\alpha$ and McDonald's ω coefficients of .85, acting with awareness subscale had Cronbach's $\alpha$ and McDonald's ω coefficients of .87, ability to describe subscale had Cronbach's $\alpha$ and McDonald's ω coefficients of .88, and being non-judgmental subscale had Cronbach's $\alpha$ of .82 and McDonald's ω coefficient of .83.

Another instrument employed to assess dispositional mindfulness, was the Mindfulness Attention and Awareness Scale (MAAS) [7]. The MAAS consists of 15 items measured on a 6-point Likert scale (1 –Almost always, to 6 –Almost never) that asks participants about how they normally go through activities in their daily lives (e.g., "I rush through activities without being really attentive to them"). Internal reliability was Cronbach's $\alpha$ of .88 and McDonald's ω coefficient of .89.

**Trait socio-cognitive mindfulness** was assessed with the Langer mindfulness Scale which assesses novelty seeking, novelty producing, and engagement (LMS) [29]. Novelty seeking is measured by items such as "I do not actively seek to learn new things". The subscale novelty producing includes items such as "I find it easy to create new and effective ideas", and the subscale engagement includes items such as "I 'get involved' in almost everything I do". Participants rate their agreement with items on a 7-point scale ranging from "1 = strongly disagree" to "7 = strongly agree". In the present research, the subscales demonstrated adequate to good internal consistencies. Novelty seeking had Cronbach's $\alpha$ and McDonald's ω coefficients of .82, novelty producing had Cronbach's $\alpha$ of .75 and McDonald's ω coefficients of .77 and engagement had Cronbach's $\alpha$ of .62 and McDonald's ω coefficients of .63.

**Encoding style** was assessed with Encoding Style Questioner (ESQ) [30]. This scale consists of 9 items evaluating the level of internal encoding. Representative items include, "For a split second from a distance, I sometimes mistake strangers for people I know", and "Sometimes when I'm driving, I see a piece of paper or a leaf being moved by the wind, and for a split-second think that it might be an animal". Participants respond to each item using a 6-point Likert type scale, ranging from "1 = Never" to "7 = Often". A high score on the ESQ reflects an internal encoding style, whereas a low score reflects an external encoding style. In the current study the ESQ had Cronbach's $\alpha$ of .84 and McDonald's ω coefficients of .85.

### Statistical plan

Data were cleaned using Microsoft Excel and analyzed with JASP version 0.17.1 [31]. The sample was randomly split into two datasets of approximately equal size and with approximately equal number of men and women: the Exploratory Factor Analysis (EFA, $N$ = 290) was performed on one half, and the Confirmatory Factor Analysis (CFA, $N$ = 282) was performed on the other half. Pearson correlations were analyzed to assess the correlations between total assessments' scores (i.e., SA, MD, MAAS, FFMQ, ESQ).

### Results

Prior to conducting the EFA and CFA, the two random samples were tested for equal distributions of Gender, Age, and Religiosity. Chi square independence test revealed that Gender distributed equally between the samples, $\chi 2(2)$ = 2.66, $p$ = .26. Similarly, t-test analyses for

independent samples confirmed equal distribution of Age, $t(570) = 1.89$, $p = .06$, as well as of religiosity, $t(522) = 0.32$, $p = .75$.

## EFA

The EFA was aimed to examine whether Synchronicity Awareness (SA) and Meaning Detection (MD) items are loaded onto two separate factors, that are distinguishable from the items and factors of mindfulness and encoding style. To this end, all items from the SA, MD, Five Facets of Mindfulness Questionnaire (FFMQ), Langer Mindfulness Scale (LMS), and Encoding Style Questionnaire (ESQ), were submitted to the EFA. Items of the Mindfulness Attention and Awareness Scale (MAAS) were excluded because they mainly overlap with the awareness facet of the FFMQ. First, Kaiser–Meyer–Olkin (KMO) test and Bartlett's test of sphericity were conducted to confirm that our data are adequate for factor analysis. Indeed, the KMO revealed and overall MSA value of .85, and Bartlett's Chi square was $\chi^2(2346) = 9831.14$, $p < .001$, thus indicating that the data are adequate for factor analysis.

Next, the EFA was conducted using a minimum residual estimation method and by applying an Oblimin rotation. The analysis yielded nine factors (factor loadings are presented in Table 1). The analysis revealed that overall, the items were loaded onto their designated factor, explaining 46% of the variance and presented a good fit ($\chi^2(1761) = 2444.74$, $p < .001$; RMSEA = .036 [90% CI: .033-.04], SRMR = .034, TLI = .875, CFI = .909). It seems, therefore, that the SA and MD items are distinguishable from the mindfulness and encoding style items, and they constitute two distinct constructs. Nevertheless, one item from each of the following scales, MD, SA, and ESQ, was not loaded onto its corresponding factor. For the LMS, three "engagement" items and one "novelty producing" item were not loaded onto their corresponding factors.

## CFA

The CFA was conducted to check the validity of the notion that SA and MS are two factors distinct from each other and from the well-established mindfulness and encoding style factors. To this end, 11 factors corresponding with SA, MD, ESQ, five factors of the FFMQ, and three factors of the LMS, were submitted to the CFA. FFMQ and LMS factors were submitted as eight first-order factors (i.e., without a latent factor) in order to better understand the interplay of SA and MD with mindfulness (factor loadings are presented in Table 2). Overall, the CFA indicated good fit, and provided the following goodness of fit indices: $\chi^2(2222) = 5527.44$, $p < .001$; RMSEA = .050 (90% CI [.047, .052]), SRMR = .069, CFI = .972, NFI = .948, GFI = .973, and TLI = .970, hence validating that SA and MD are distinguishable from the factors of mindfulness and encoding style.

## Correlation analysis

Finally, after establishing that SA and MD are two standalone constructs, we turned to examine the correlations between the different constructs in our study. This analysis was carried on the entire sample ($N = 572$) and its coefficients are presented in Table 3. It was found that encoding style was mainly negatively associated with MAAS and FFMQ scores, indicating that mindfulness was associated with external encoding style. More important, Table 3 reveals that SA nor MD were correlated with age, but both were positively correlated with religiosity, indicating that participants reporting to be more religious also reported elevated rates of synchronicity awareness and meaning detection.

As for the trait-like factors, SA exhibited a medium to strong positive association with MD, but medium to weak association with encoding style, all factors of Langer's Mindfulness Scale,

**Table 1. Factor loadings and uniqueness of the research items.**

| Item | F1 | F2 | F3 | F4 | F5 | F6 | F7 | F8 | F9 | Uniqueness |
|------|-----|-----|-----|-----|-----|-----|-----|-----|-----|------------|
| MD1 | 0.784 | | | | | | | | | 0.421 |
| MD5 | 0.719 | | | | | | | | | 0.336 |
| MD9 | 0.716 | | | | | | | | | 0.429 |
| MD6 | 0.682 | | | | | | | | | 0.395 |
| MD3 | 0.662 | | | | | | | | | 0.485 |
| MD2 | 0.650 | | | | | | | | | 0.592 |
| MD12 | 0.616 | | | | | | | | | 0.490 |
| MD11 | 0.597 | | | | | | | | | 0.426 |
| MD13 | 0.574 | | | | | | | | | 0.634 |
| MD10 | 0.570 | | | | | | | | | 0.526 |
| MD4 | 0.541 | | | | | | | | | 0.436 |
| MD8 | 0.521 | | | | | | | | | 0.579 |
| LMS-NS5 | | 0.711 | | | | | | | | 0.446 |
| LMS-NS3 | | 0.644 | | | | | | | | 0.447 |
| LMS-NS1 | | 0.622 | | | | | | | | 0.520 |
| LMS-NP4 | | 0.612 | | | | | | | | 0.385 |
| LMS-NP5 | | 0.607 | | | | | | | | 0.568 |
| LMS-NS2 | | 0.604 | | | | | | | | 0.553 |
| LMS-NP3 | | 0.509 | | | | | | | | 0.597 |
| LMS-NS4 | | 0.484 | | | | | | | | 0.672 |
| LMS-NP1 | | 0.478 | | | | | | | | 0.547 |
| LMS-E4 | | 0.464 | | | | | | | | 0.654 |
| ESQ4 | | | 0.736 | | | | | | | 0.483 |
| ESQ7 | | | 0.674 | | | | | | | 0.518 |
| ESQ3 | | | 0.667 | | | | | | | 0.486 |
| ESQ9 | | | 0.602 | | | | | | | 0.499 |
| ESQ2 | | | 0.568 | | | | | | | 0.589 |
| ESQ5 | | | 0.540 | | | | | | | 0.580 |
| ESQ6 | | | 0.470 | | | | | | | 0.642 |
| ESQ1 | | | 0.454 | | | | | | | 0.760 |
| FFMQ-Aw2 | | | | 0.763 | | | | | | 0.307 |
| FFMQ-Aw3 | | | | 0.701 | | | | | | 0.384 |
| FFMQ-Aw5 | | | | 0.687 | | | | | | 0.427 |
| FFMQ-Aw4 | | | | 0.654 | | | | | | 0.481 |
| FFMQ-Aw1 | | | | 0.442 | | | | | | 0.562 |
| FFMQ-Ds1 | | | | | 0.829 | | | | | 0.294 |
| FFMQ-Ds2 | | | | | 0.739 | | | | | 0.390 |
| FFMQ-Ds3 | | | | | 0.627 | | | | | 0.407 |
| FFMQ-Ds4 | | | | | 0.591 | | | | | 0.532 |
| FFMQ-Ds5 | | | | | 0.570 | | | | | 0.476 |
| SA7 | | | | | | 0.662 | | | | 0.521 |
| SA9 | | | | | | 0.611 | | | | 0.610 |
| SA6 | | | | | | 0.570 | | | | 0.611 |
| SA1 | | | | | | 0.555 | | | | 0.542 |
| SA4 | | | | | | 0.501 | | | | 0.585 |
| SA8 | | | | | | 0.445 | | | | 0.664 |
| SA2 | | | | | | 0.418 | | | | 0.758 |

*(Continued)*

**Table 1.** (Continued)

| Item | F1 | F2 | F3 | F4 | F5 | F6 | F7 | F8 | F9 | Uniqueness |
|---|---|---|---|---|---|---|---|---|---|---|
| SA5 | | | | | | 0.413 | | | | 0.672 |
| FFMQ-NR2 | | | | | | | 0.790 | | | 0.278 |
| FFMQ-NR5 | | | | | | | 0.735 | | | 0.431 |
| FFMQ-NR4 | | | | | | | 0.707 | | | 0.441 |
| FFMQ-NR3 | | | | | | | 0.686 | | | 0.487 |
| FFMQ-NR1 | | | | | | | 0.580 | | | 0.621 |
| FFMQ-NJ4 | | | | | | | | 0.716 | | 0.374 |
| FFMQ-NJ3 | | | | | | | | 0.666 | | 0.399 |
| FFMQ-NJ1 | | | | | | | | 0.642 | | 0.436 |
| FFMQ-NJ5 | | | | | | | | 0.588 | | 0.610 |
| FFMQ-NJ2 | | | | | | | | 0.485 | | 0.673 |
| FFMQ-Ob1 | | | | | | | | | 0.637 | 0.495 |
| FFMQ-Ob4 | | | | | | | | | 0.585 | 0.537 |
| FFMQ-Ob2 | | | | | | | | | 0.570 | 0.617 |
| FFMQ-Ob3 | | | | | | | | | 0.479 | 0.688 |
| MD7 | | | | | | | | | | 0.664 |
| SA3 | | | | | | | | | | 0.656 |
| ESQ8 | | | | | | | | | | 0.758 |
| LMS-NP2 | | | | | | | | | | 0.897 |
| LMS-E2 | | | | | | | | | | 0.824 |
| LMS-E1 | | | | | | | | | | 0.752 |
| LMS-E3 | | | | | | | | | | 0.698 |

*Note*. SA- Synchronicity Awareness; MD- Meaning Detection; FFMQ- Five Facets of Mindfulness Questionnaire; LMS- Langer Mindfulness Scale; ESQ- Encoding Style Questionnaire; Aw- Acting with awareness; Ds- Describe; NR- Acting without reacting; NJ- Acting without judgment; Ob- Observing; NS- Novelty seeking; NP- Novelty producing; E- engagement.

and the FFMQ's facets of Observing and Describing. Similar associations were observed for MD, with the exception of MD being negatively associated with the FFMQ facet of Acting without judging. To sum, these associations indicate that both SA and MD may be characterized with participants' tendency to interpret environmental cues as preexisting experiences (i.e., internal encoding style), with participants' increased interoceptive awareness and with the ability to describe these internal feelings, as well as with increased mindful thinking. Additional analyses are presented in S1 Appendix.

## Discussion

The purpose of the present study was twofold: to further validating the new Synchronicity Awareness and Meaning Detection (SAMD) questionnaire, and to explore the manner in which the concept of synchronicity is related to the concept of mindfulness. Overall, our findings support our predictions and indicate that synchronicity is intimately related to some core features of mindfulness.

Specifically, as predicted, synchronicity was associated with inward-directed attention, which is a core facet of mindfulness [12], as well as with the Langerian concept of mindfulness, which emphasizes the role of meaningful interactions in everyday experiences [9]. That is, participants with higher SA and MD scores were characterized as more observing, novelty-seeking, novelty-producing, and with the tendency to be engaged with their actions. The observed

**Table 2. CFA factor loadings and residual variance of the research items.**

| Factor | Item | Loading | Residual Variance |
|---|---|---|---|
| Synchronicity Awareness | SA1 | 0.65 | 0.57 |
| | SA2 | 0.58 | 0.66 |
| | SA3 | 0.73 | 0.47 |
| | SA4 | 0.50 | 0.75 |
| | SA5 | 0.66 | 0.57 |
| | SA6 | 0.60 | 0.64 |
| | SA7 | 0.61 | 0.63 |
| | SA8 | 0.56 | 0.69 |
| | SA9 | 0.56 | 0.69 |
| Meaning Detection | MD1 | 0.76 | 0.43 |
| | MD2 | 0.65 | 0.57 |
| | MD3 | 0.71 | 0.49 |
| | MD4 | 0.64 | 0.59 |
| | MD5 | 0.74 | 0.45 |
| | MD6 | 0.75 | 0.44 |
| | MD7 | 0.61 | 0.63 |
| | MD8 | 0.68 | 0.54 |
| | MD9 | 0.68 | 0.54 |
| | MD10 | 0.69 | 0.52 |
| | MD11 | 0.76 | 0.43 |
| | MD12 | 0.66 | 0.57 |
| | MD13 | 0.62 | 0.62 |
| Encoding Style | ESQ1 | 0.50 | 0.75 |
| | ESQ2 | 0.69 | 0.52 |
| | ESQ3 | 0.66 | 0.57 |
| | ESQ4 | 0.79 | 0.38 |
| | ESQ5 | 0.63 | 0.61 |
| | ESQ6 | 0.64 | 0.59 |
| | ESQ7 | 0.73 | 0.47 |
| | ESQ8 | 0.65 | 0.57 |
| | ESQ9 | 0.73 | 0.47 |
| LMS: Novelty Seeking | LMS-NS1 | 0.84 | 0.30 |
| | LMS-NS2 | 0.80 | 0.37 |
| | LMS-NS3 | 0.78 | 0.40 |
| | LMS-NS4 | 0.64 | 0.59 |
| | LMS-NS5 | 0.67 | 0.55 |
| LMS: Novelty Producing | LMS-NP1 | 0.24 | 0.94 |
| | LMS-NP2 | 0.75 | 0.43 |
| | LMS-NP3 | 0.73 | 0.46 |
| | LMS-NP4 | 0.87 | 0.24 |
| | LMS-NP5 | 0.65 | 0.57 |
| LMS: Engagement | LMS-E1 | 0.45 | 0.80 |
| | LMS-E2 | 0.67 | 0.56 |
| | LMS-E3 | 0.63 | 0.61 |
| | LMS-E4 | 0.70 | 0.51 |

(*Continued*)

**Table 2.** (*Continued*)

| Factor | Item | Loading | Residual Variance |
|---|---|---|---|
| FFMQ: Observing | Observe1 | 0.87 | 0.24 |
| | Observe2 | 0.75 | 0.44 |
| | Observe3 | 0.67 | 0.55 |
| | Observe4 | 0.73 | 0.47 |
| FFMQ: Acting with Awareness | Aware1 | 0.80 | 0.34 |
| | Aware2 | 0.62 | 0.53 |
| | Aware3 | 0.66 | 0.48 |
| | Aware4 | 0.72 | 0.41 |
| | Aware5 | 0.81 | 0.27 |
| FFMQ: Describe | Desc1 | 0.90 | 0.19 |
| | Desc2 | 0.85 | 0.27 |
| | Desc3 | 0.89 | 0.22 |
| | Desc4 | 0.66 | 0.57 |
| | Desc5 | 0.83 | 0.31 |
| FFMQ: Acting without Judgement | NJ1 | 0.82 | 0.33 |
| | NJ2 | 0.45 | 0.80 |
| | NJ3 | 0.87 | 0.25 |
| | NJ4 | 0.86 | 0.27 |
| | NJ5 | 0.57 | 0.68 |
| FFMQ: Acting without Reacting | NR1 | 0.62 | 0.61 |
| | NR2 | 0.88 | 0.23 |
| | NR3 | 0.69 | 0.53 |
| | NR4 | 0.68 | 0.54 |
| | NR5 | 0.78 | 0.39 |

*Note.* All loadings significantly differed from 0 at $p < .001$.

association of synchronicity with the Langerian concept of mindfulness is intriguing. Langer's mindfulness conceptualization posits that mindfulness is not merely about being present-focused or non-judgmental, but also about engaging meaningfully with one's environment and experiences.

This view emphasizes the relevance of novelty and contextual variability, suggesting that mindful individuals are those who remain open to new information, consistently seek to explore novel perspectives, and avoid entrapment in rigid cognitive schemas. This notion aligns with the characteristics of participants with high SAMD scores, further cementing the connection between synchronicity and mindfulness. The fact that synchronicity experiences entail being attuned to random and unexpected cues in a complex, stimulus-rich environment, while concurrently maintaining a connection and awareness of one's inner world, implies that synchronicity may provide a rich and unique context for exploring mindfulness concepts.

Another important finding in the study was, as predicted, the association between SAMD scores and Lewicki's encoding style [30]. Critically, participants with higher SAMD scores tended to demonstrate a more top-down (i.e., internal encoding style) of everyday experiences. Such a top-down encoding style implies a tendency to process information in a manner driven by cognitive constructs, prior knowledge or expectancy. Such a notion could entail one's attributing meaning to their experiences based on their pre-existing frameworks or conceptual schemas, rather than being purely influenced by the immediate sensory input. This notion is

**Table 3. Pearson correlation coefficients between the study variables.**

| Variable | Mean (SD) $\alpha_c$ | 1 | 2 | 3 | 4 | 5 | 6 | 7 | 8 | 9 | 10 | 11 | 12 | 13 | 14 |
|---|---|---|---|---|---|---|---|---|---|---|---|---|---|---|---|
| 1. Age | 42.95 (14.03) | | | | | | | | | | | | | | |
| 2. Religiosity | 1.79 (1.16) | .117** | | | | | | | | | | | | | |
| 3. SA | 2.20 (0.83) .81 | .062 | .181*** | | | | | | | | | | | | |
| 4. MD | 54.65 (13.15) .91 | .007 | .316*** | .548*** | | | | | | | | | | | |
| 5. ESQ | 3.01 (0.94) .84 | -.137** | .137** | .335*** | .332*** | | | | | | | | | | |
| 6. MAAS | 3.83 (0.77) .88 | .246*** | -.008 | -.077 | -.071 | -.310*** | | | | | | | | | |
| 7. LMS-T | 4.88 (0.78) .85 | .105* | .145*** | .253*** | .345*** | .015 | .337*** | | | | | | | | |
| 8. LMS-NS | 5.17 (0.98) .82 | .108** | .120** | .232*** | .371*** | .052 | .212*** | .853*** | | | | | | | |
| 9. LMS-NP | 4.30 (1.01) .75 | .060 | .117** | .208*** | .297*** | -.002 | .286*** | .858*** | .615*** | | | | | | |
| 10. LMS-E | 5.24 (0.91) .63 | .088* | .111* | .157*** | .122** | -.023 | .329*** | .658*** | .354*** | .356*** | | | | | |
| 11. FFMQ-Ob | 3.65 (0.73) .85 | .206*** | .146*** | .338*** | .457*** | .174*** | .172*** | .406*** | .351*** | .318*** | .303*** | | | | |
| 12. FFMQ-Aw | 3.27 (0.79) .87 | .206*** | .015 | -.043 | -.055 | -.239*** | .778*** | .331*** | .231*** | .269*** | .309*** | .129** | | | |
| 13. FFMQ-NJ | 3.08 (0.82) .82 | .205*** | -.048 | -.059 | -.148*** | -.237*** | .405*** | .159*** | .070 | .155*** | .166*** | .034 | .375*** | | |
| 14. FFMQ-NR | 3.02 (0.79) .85 | .142*** | -.005 | .044 | .045 | -.108** | .298*** | .188*** | .192*** | .200*** | .025 | .078 | .250*** | .380*** | |
| 15. FFMQ-Ds | 3.30 (0.85) .88 | .147*** | .127** | .126** | .111** | -.140*** | .440*** | .399*** | .275*** | .354*** | .333*** | .227*** | .423*** | .335*** | .263*** |

*Note*. SA- Synchronicity Awareness; MD- Meaning Detection; FFMQ- Five Facets of Mindfulness Questionnaire; LMS- Langer Mindfulness Scale; ESQ- Encoding Style Questionnaire; Aw- Acting with awareness; Ds- Describe; NR- Acting without reacting; NJ- Acting without judgment; Ob- Observing; T- Total score; NS- Novelty seeking; NP- Novelty producing; E- engagement.

* $p < .05$

** $p < .01$

*** $p < .001$

important for understanding the interrelations between synchronicity and mindfulness, because previous studies have shown that mindfulness was associated with an external, rather than internal, encoding style [14], as was indeed evident in our findings.

The categorization of mindfulness as either a top-down or bottom-up process remains a topic of ongoing debate. However, certain attention and perception-related studies present persuasive arguments that mindfulness may foster top-down processing [32, 33]. In the context of emotion regulation, it has been proposed that mindfulness might initially function as a top-down, ad-hoc regulatory process for meditation novices [34]. However, with continued practice, it may transition into a bottom-up regulatory process. Viewing synchronicity through this perspective suggests it might be characterized by a distinct cognitive style, yet is triggered by unique, ad-hoc circumstances. Consequently, SAMD scores were correlated with both high levels of mindfulness and an internal encoding style, while high mindfulness scores were simultaneously associated with an external encoding style.

Interestingly, in contrast to our prediction, synchronicity was not associated with mindful awareness, as assessed by the MAAS [7] and the Acting with awareness facet of the FFMQ [8]. This suggest that synchronicity may not be characterized with outwards-directing attention or thoughtful actions; alternately, synchronicity may reflect a more intrinsic attentional process, in which an experience has no value by itself but rather is automatically defined and contextualized by current sensations, feelings, thoughts, and memories [11]. In line with this notion, it has been argued that the relationship between mindfulness and memory may not be mediated by improved outwards-oriented attention, but has more to do with decision-making processes affected by internal cues. Synchronicity, therefore, entails maintaining a receptive mindset toward unexpected and unexplained experiences in life [3, 4] and the interpretation of these experiences through internal ad-hoc cues.

Indeed, some support for this notion may be found in the negative association between MD and Acting without judgment scores, which indicated that higher tendency for meaning detection was accompanied by one's being judgmental towards the experience. Hence, maintaining synchronicity awareness entails holding an appraising and interpreting perspective of the current experience, in the sense of one's evaluation of the relevance and role of the situation for his or her life narrative.

From our perspective, the connection between SAMD and mindfulness could be based on two cognitive processes. Firstly, research indicates that mindfulness practice may enhance visual processing capabilities. Studies have demonstrated that after practicing mindfulness, participants were able to complete visual perception tasks faster and more accurately compared to controls [35, 36]. This suggests that mindfulness may foster more effective processing of environmental cues, rendering them more salient; this, in turn, could lead to elevated synchronicity awareness.

Secondly, mindfulness practice has been argued to engage meta-cognitive processes [37, 38], which might partially drive synchronicity awareness. This concept aligns with previous findings [12], suggesting that mindful individuals could more accurately incorporate meta-cognitive processes, such as the episodic familiarity evoked by fluent processing, to determine whether they had previously studied a target word. Therefore, during a given experience, mindfulness might promote enhanced awareness to thoughts and feelings and facilitate the retrieval of information from episodic memory, thus evoking synchronicity awareness. However, considering our study is of a correlational nature, further research is required to better understand the cognitive processes that connect mindfulness and synchronicity.

## Limitations and suggestions for future research

The present study has two main limitations that should be taken into consideration. First, our sample consisted of English-speaking Westerners, the majority of whom identified as Caucasians. Because the mindfulness questionnaires employed in this study were conceived by Western scholars and articulated in English, there may be a plausible confinement to the validity of our findings strictly to this demographic. Consequently, a cautious approach must be taken in extending these insights to a more universal human experience. Indeed, this caveat has been previously discussed [39], with the notion that mindfulness questionnaires may be culturally biased to some extent thus should be prudently interpreted. It should be noted, however, that our sample did not involve undergraduate students, but rather a community sample and hence more likely to represent the wide population. Furthermore, a comparison between Caucasians and non-Caucasians revealed no significant differences in any of the study variables, but in Age (Caucasians were 6 years older in average) and Religiosity (Caucasians reported to be somewhat less religious). Therefore although we cannot generalize the findings to different ethnicities than Caucasians, it seems that the experience of synchronicity may be an essential human experience and not limited to the current sample demographics. Nonetheless, Future studies should also include cultural variables in order to shed light on the question.

A second limitation lies in the correlational nature of our study. Because mindfulness and synchronicity were not cultivated but assessed, we cannot conclude that mindfulness increases synchronicity awareness. Nevertheless, in this study we sought to explore whether mindfulness and synchronicity awareness are distinct constructs as well as the interplay between them, hence investigating the impact of mindfulness on synchronicity awareness was not in the scope of the present study. It bears emphasizing that unveiling the potential causal linkage between mindfulness and synchronicity is important for better understanding the cognitive

mechanisms of mindfulness and may entail some real-world implications, hence should be addressed in future studies.

Altogether, we suggest that mindfulness and synchronicity are distinct-yet-related constructs. Their intersection may reside in some bottom-up perceptual processes, as increased sensitivity to cues or broad perceptual span. Alternatively, their intersection may reside in similar top-down meta-cognitive processes occurring at the fringes of consciousness. Future studies may illuminate the underlying commonalities of mindfulness and synchronicity by comparing participants with high compared to low synchronicity awareness in perceptual capabilities or in tasks promoting processing at the fringes of consciousness. Diary or Ecological Momentary Assessment studies may also be employed to reveal the manner to which mindfulness and synchronicity awareness are intertwined. Understanding the conditions which facilitate or hinder synchronicity experiences may enrich not only the field of mindfulness, but also the related fields of spirituality, positive psychology, and psychotherapy. Future investigations in this research trajectory should prioritize encompassing a more diverse participant pool to foster inclusivity and global perspective, thereby nurturing a well-rounded comprehension of the phenomena in question.

Overall, despite its limitations, this study extends existing literature of clinical reports and case studies on the phenomenon of synchronicity by taking a step further to provide possible directions to better understand the cognitive mechanisms underlying the experience of synchronicity. This study also lay the ground for a new direction of research in the field of mindfulness and provides a new perspective on the manner to which mindfulness may shape our perception of daily experiences.

## Supporting information

**S1 Appendix.**
(DOCX)

## Author Contributions

**Conceptualization:** Eyal Rosenstreich, Pninit Russo-Netzer, Tamar Ickeson.

**Writing – original draft:** Eyal Rosenstreich, Pninit Russo-Netzer, Tamar Ickeson.

**Writing – review & editing:** Eyal Rosenstreich, Pninit Russo-Netzer, Tamar Ickeson.

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
