## [Decision Letter · Decision Letter 0]

6 Mar 2024

PONE-D-23-43024Beyond coincidence: An Investigation of the Interplay Between Synchronicity Awareness and the Mindful State.PLOS ONE

Dear Dr. Rosenstreich,

Thank you for submitting your manuscript to PLOS ONE. After careful consideration, we feel that it has merit but does not fully meet PLOS ONE’s publication criteria as it currently stands. Therefore, we invite you to submit a revised version of the manuscript that addresses the points raised during the review process.

 The Reviewers both indicate that the study is novel and useful for growing the research field of mindfulness. However, they also raised major and minor issues that the authors are invited to address with a substantial revision.

We look forward to receiving your revised manuscript.

Kind regards,

Vilfredo De Pascalis

Academic Editor

PLOS ONE

Journal Requirements:

Whilst you may use any professional scientific editing service of your choice, PLOS has partnered with both American Journal Experts (AJE) and Editage to provide discounted services to PLOS authors. Both organizations have experience helping authors meet PLOS guidelines and can provide language editing, translation, manuscript formatting, and figure formatting to ensure your manuscript meets our submission guidelines. To take advantage of our partnership with AJE, visit the AJE website (http://aje.com/go/plos) for a 15% discount off AJE services. To take advantage of our partnership with Editage, visit the Editage website (www.editage.com) and enter referral code PLOSEDIT for a 15% discount off Editage services. If the PLOS editorial team finds any language issues in text that either AJE or Editage has edited, the service provider will re-edit the text for free.

3. In the online submission form, you indicated that Data are available from the authors (contact via email: eyal@pac.ac.il) without restrictions.

Additional Editor Comments:

The Reviewers both indicate that the study is novel and useful for growing the research field of mindfulness. However, they also raised major and minor issues that the authors are invited to address with a substantial revision.

Reviewers' comments:

Reviewer's Responses to Questions

**Comments to the Author**

1. Is the manuscript technically sound, and do the data support the conclusions?

Reviewer #1: Yes

Reviewer #2: Yes

2. Has the statistical analysis been performed appropriately and rigorously? 

Reviewer #1: Yes

Reviewer #2: Yes

3. Have the authors made all data underlying the findings in their manuscript fully available?

Reviewer #1: No

Reviewer #2: No

4. Is the manuscript presented in an intelligible fashion and written in standard English?

Reviewer #1: No

Reviewer #2: Yes

5. Review Comments to the Author

Reviewer #1: PLOS review 2024 2 26

Here the authors sought to validate a novel scale (SAMD) aimed at characterizing the mechanisms underlying synchronicity awareness, a ubiquitous yet understudied aspect of conscious experience, and probe its relation to mindfulness.

Strengths:

The authors make a good case for the importance of synchronicity for mental health and wellbeing , i.e. “ it may be suggested that being aware of our present-moment experiences enhances our ability to facilitate a dialogue between our internal world and external environment and to make sense of random and unexpected information and events in individuals' everyday lives in natural settings.

“

This study is novel and will be useful for the growing field of mindfulness studies.

The study is well powered, and the analyses seem clear, justified and well-performed.

Weaknesses:

I have several issues with the manuscript in its current form that should be addressed.

Furthermore, it is hampered by a lack of clarity and care in the writing and should undergo an in-depth round of editing before resubmission.

General points:

The study should be framed as one of bringing an aspect of mindfulness, “synchronicity awareness” in relation to other more studied aspects of the greater construct of mindfulness. Otherwise the rationale is not immediately clear from the abstract.

The reason for a dominantly caucasian sample is unclear (given the highly diverse populations available in US and UK), and will limit generalizability. I would strongly recommend recruiting more non-caucasian participants.

The amount of compensation received by participants is not mentioned.

Points for improvement in writing (due to the unfortunately high number of incidences of typos, grammatical errors, and awkward or disorganized composition, I could not provide a full list. Please revise considerably before submitting):

P.2, L.3 should read” it has scarcely been examined scientifically and little is known…”

P.2, L.6 should read “engaged in an online survey…”

P.3, L. 2 should read “mindful state is characterized”

P.3. L.5-6 this sentence is awkward , should read something like” mindfulness has been conceived as an acquired skill, and as a trait-like construct”

P.3, L.8 should read “multifaceted construct”

P.3, L.12, 14, should read “curious” and innovative thought”

P.3, L.20, The sentence starting with “for example, mindfulness” needs to be rewritten for consistency of tense (which manifested), and I assume you mean “processing and evoking ‘a feeling of memory’ during…”?

P.5, L.18, the sentence beginning with “given that…” is a disorganized, run-on sentence. Please simplify or break up into smaller sentences.

P.6 L.19, The last sentence is extremely vague and unnecessary. Potential meeting points and relationships between what?

…….

P.9, L.19, should read “Trait socio-cognitive”, i.e. no “A” is necessary.

Reviewer #2: The study aimed to achieve two primary objectives: (1) to further validate the SAMD questionnaire in a different population, and (2) to investigate whether awareness synchrony represents a distinct construct from mindfulness. The questions posed in the study are intriguing, and the employed methods, encompassing data analyses, are sound. I have only a few comments:

-Given that one of the primary goals was to validate the SAMD questionnaire, it would be beneficial for the authors to conduct additional analyses to assess its reliability. For instance, exploring two subscales correlation, item-subscale correlation, and inter-item correlation could offer more insights. They might even consider dividing the results section into two subsections, with one dedicated to SAMD validation and the second to construct validity. Additionally, including Cronbach's alpha in the results section rather than the methods section would enhance clarity.

-It is unclear why the MAAS was not included in the EFA. While I acknowledge the potential redundancy with the FFMQ, I don’t get how it is an issue to test whether SAMD items load on two distinguishable factors, independent of MAAS.

-The decision to provide loading factors for only one of the factors in the EFA raises questions. It would be valuable to observe how different items load on various factors, providing a more comprehensive understanding of the questionnaire's structure.

-The sentence, "Nevertheless, one item from each of the following scales, MD, SA, and ESQ, was loaded onto its corresponding factor. For the LMS, three 'engagement' items and one 'novelty producing' item were loaded onto their corresponding factors," lacks clarity regarding which corresponding factor is being referred to. Including all loading factors would enhance the clarity of this statement.

-It would also be interesting to provide the correlations between factors in both the EFA and CFA as they would also inform the relationships between the identified factors.

Typo: Introduction, L2 : "This mindful state is characterized" not "This mindful state characterized"

6. PLOS authors have the option to publish the peer review history of their article (what does this mean?). If published, this will include your full peer review and any attached files.

Reviewer #1: **Yes: **Leonardo Christov-Moore

Reviewer #2: No

---

## [Author Response · Author response to Decision Letter 0]

10 Apr 2024

We wish to thank both reviewers for their insightful comments. We have made a number of changes to address all the issues raised by the reviewers. The paper is considerably stronger as a result. Below we detail the changes we made in blue.

Reviewer #1: 

The manuscript was submitted to language editing. Corrections were made throughout the manuscript.

1. The study should be framed as one of bringing an aspect of mindfulness, “synchronicity awareness” in relation to other more studied aspects of the greater construct of mindfulness. Otherwise the rationale is not immediately clear from the abstract.

We thank Reviewer 1 for this insightful comment. We have completely rewritten the abstract and we believe it now presents the rationale more clearly. 

2. The reason for a dominantly caucasian sample is unclear (given the highly diverse populations available in US and UK), and will limit generalizability. I would strongly recommend recruiting more non-caucasian participants.

We thank Reviewer 1 for this comment. Although the Prolific holds a highly diverse pool of participants, unfortunately most of the people choosing to take part in our study identified as Caucasians. We do agree that our ability to generalize the findings has been limited due to sample characteristics, and we now address this issue in the discussion (see P. 18 top). Nevertheless, following this comment we examined group differences between Caucasians and non-Caucasians, and found that these groups differed solely in Age (Caucasians were 6 years older in average) and Religiosity (Caucasians reported to be less religious). No differences were observed in any of the critical variables of the study. 

3. The amount of compensation received by participants is not mentioned. Thank you for your attention. This was added. Please see page 8.

4. Points for improvement in writing (due to the unfortunately high number of incidences of typos, grammatical errors, and awkward or disorganized composition, I could not provide a full list. Please revise considerably before submitting):

Thank you for your attention. Indeed, many typos and grammatical errors were found and hopefully we got them all now. 

P.2, L.3 should read” it has scarcely been examined scientifically and little is known…”

This was corrected. Thank you.

P.2, L.6 should read “engaged in an online survey…” Done

P.3, L. 2 should read “mindful state is characterized” Done

P.3. L.5-6 this sentence is awkward , should read something like” mindfulness has been conceived as an acquired skill, and as a trait-like construct” Done

P.3, L.8 should read “multifaceted construct” Done

P.3, L.12, 14, should read “curious” and innovative thought” Done

P.3, L.20, The sentence starting with “for example, mindfulness” needs to be rewritten for consistency of tense (which manifested), and I assume you mean “processing and evoking ‘a feeling of memory’ during…”? Done

P.5, L.18, the sentence beginning with “given that…” is a disorganized, run-on sentence. Please simplify or break up into smaller sentences. Done

P.6 L.19, The last sentence is extremely vague and unnecessary. Potential meeting points and relationships between what? Done

P.9, L.19, should read “Trait socio-cognitive”, i.e. no “A” is necessary. Done

Reviewer #2: 

1. Given that one of the primary goals was to validate the SAMD questionnaire, it would be beneficial for the authors to conduct additional analyses to assess its reliability. For instance, exploring two subscales correlation, item-subscale correlation, and inter-item correlation could offer more insights. 

We now present these analyses in the Appendix. See Tables A and B.

2. They might even consider dividing the results section into two subsections, with one dedicated to SAMD validation and the second to construct validity. 

We thank Reviewer 2 for this helpful suggestion. We tried dividing the Results section, but we felt that the outcome was somewhat artificial because the construct validity is an inherent part of the whole validation process. Therefore, we decided to keep the Results in their original format.

3. Additionally, including Cronbach's alpha in the results section rather than the methods section would enhance clarity.

Cronbach’s alphas are now presented in Table 2.

4. It is unclear why the MAAS was not included in the EFA. While I acknowledge the potential redundancy with the FFMQ, I don’t get how it is an issue to test whether SAMD items load on two distinguishable factors, independent of MAAS.

Thank you for this insightful comment. The MAAS was highly correlated with FFMQ’s ‘Acting with awareness’ (Pearson’s r = 0.78). Because EFA is basically a set of multiple regression analyses with are sensitive to multicollinearity, we decided to exclude the MAAS from this analysis. Indeed, scholars have argued that multicollinearity becomes an issue in EFA when it reaches .9 and above, but we decided to approach it with extreme care because the MAAS and FFMQ’s Awareness are not merely highly correlated but are basically the same thing (FFMQ’s Awareness consists of MAAS items). 

It should be noted that we examined Reviewer 2’s suggestion, and the analysis revealed better fit indices and is preferable from a purely statistic point of view (e.g., RMSEA = .030 instead of .036; TLI = .904 compared to .875 etc.), but from a methodological point of view it is wrong; Most of the variance of the MAAS has already been accounted for by the FFMQ’s Awareness, resulting in a combined MAAS-FFMQ’s Awareness combination and a few leftovers that do not account for any variance. 

5. The decision to provide loading factors for only one of the factors in the EFA raises questions. It would be valuable to observe how different items load on various factors, providing a more comprehensive understanding of the questionnaire's structure.

We apologize, but there seem to be a somewhat misunderstanding. All factor loadings are presented in Table 1. That is, loadings for all 9 factors are presented. We are not sure whether Reviewer 2 received a partial Table 1 or whether we haven't fully understood the reviewer's intention in Comment 5. We will be glad to clarify and to complete any missing information.

6. The sentence, "Nevertheless, one item from each of the following scales, MD, SA, and ESQ, was loaded onto its corresponding factor. For the LMS, three 'engagement' items and one 'novelty producing' item were loaded onto their corresponding factors," lacks clarity regarding which corresponding factor is being referred to. Including all loading factors would enhance the clarity of this statement.

Indeed, there was a typo in this sentence, with the word “not” accidentally omitted, thus entirely changing the meaning of the sentence Please see the corrected sentence on P. 12-13.

7. It would also be interesting to provide the correlations between factors in both the EFA and CFA as they would also inform the relationships between the identified factors.

This is an interesting insight. Thank you. We checked these correlations, and they were extremely high, suggesting both sets of factors to be practically identical. These correlations are now presented in the Appendix. Please see Tables C and D. 

8. Typo: Introduction, L2 : "This mindful state is characterized" not "This mindful state characterized" Done

---

## [Decision Letter · Decision Letter 1]

17 Jun 2024

PONE-D-23-43024R1Beyond coincidence: An Investigation of the Interplay Between Synchronicity Awareness and the Mindful State.PLOS ONE

Dear Dr. Rosenstreich,

Thank you for submitting your manuscript to PLOS ONE. After careful consideration, we feel that it has merit but does not fully meet PLOS ONE’s publication criteria as it currently stands. Therefore, we invite you to submit a revised version of the manuscript that addresses the points raised during the review process. The authors are invited to address the submitted comment of Reviewer 2 and resubmit the revised manuscript for acceptance evaluation.

We look forward to receiving your revised manuscript.

Kind regards,

Vilfredo De Pascalis

Academic Editor

PLOS ONE

Journal Requirements:

Additional Editor Comments:

The authors are invited to address the submitted comment of Reviewer 2 and resubmit the revised manuscript for acceptance evaluation.

Reviewers' comments:

Reviewer's Responses to Questions

**Comments to the Author**

1. If the authors have adequately addressed your comments raised in a previous round of review and you feel that this manuscript is now acceptable for publication, you may indicate that here to bypass the “Comments to the Author” section, enter your conflict of interest statement in the “Confidential to Editor” section, and submit your "Accept" recommendation.

Reviewer #1: All comments have been addressed

Reviewer #2: (No Response)

2. Is the manuscript technically sound, and do the data support the conclusions?

Reviewer #1: Yes

Reviewer #2: Yes

3. Has the statistical analysis been performed appropriately and rigorously? 

Reviewer #1: Yes

Reviewer #2: Yes

4. Have the authors made all data underlying the findings in their manuscript fully available?

Reviewer #1: Yes

Reviewer #2: Yes

5. Is the manuscript presented in an intelligible fashion and written in standard English?

Reviewer #1: Yes

Reviewer #2: Yes

6. Review Comments to the Author

Reviewer #1: (No Response)

Reviewer #2: The authors have addressed all my comments, except for the fifth one regarding loading factors that was not clear. EFA provides a loading factor of each item on each latent variable. The authors only provided loading factors on the designated latent variable (as in CFA). I was wondering why is that ? While I understand the decision to streamline the presentation, providing loading factors for each item on each latent variable could offer a richer understanding of the relationship between SA/MD items and mindfulness/encoding style.

7. PLOS authors have the option to publish the peer review history of their article (what does this mean?). If published, this will include your full peer review and any attached files.

Reviewer #1: No

Reviewer #2: **Yes: **Laurie Geers

---

## [Author Response · Author response to Decision Letter 1]

1 Jul 2024

We thank Reviewer 2 for clarifying this issue. We now present EFA factor loadings in Table 1 and CFA factor loadings in Table 2. See pages 12 and 26.

---

## [Editor Report · Decision Letter 2]

3 Jul 2024

Beyond coincidence: An Investigation of the Interplay Between Synchronicity Awareness and the Mindful State.

PONE-D-23-43024R2

Dear Dr. Rosenstreich,

We’re pleased to inform you that your manuscript has been judged scientifically suitable for publication and will be formally accepted for publication once it meets all outstanding technical requirements.

Kind regards,

Vilfredo De Pascalis

Academic Editor

PLOS ONE

Additional Editor Comments (optional):

The authors have properly addressed the suggested issue by Reviewer 2. Thus, the manuscript can be accepted for publication.
---

## [Editor Report · Acceptance letter]

8 Jul 2024

PONE-D-23-43024R2 

PLOS ONE

Dear Dr. Rosenstreich, 

I'm pleased to inform you that your manuscript has been deemed suitable for publication in PLOS ONE. Congratulations! Your manuscript is now being handed over to our production team.

Kind regards, 

on behalf of

Prof. Vilfredo De Pascalis 

Academic Editor

PLOS ONE